# 1 Reconstructing Global Monthly Ocean Dissolved Oxygen

- 2 (1960 2023) to Nearly 6000 m Depth Using Bayesian
- **3 Ensemble Machine Learning**
- Mingyu Han<sup>1</sup>, Yuntao Zhou<sup>1\*</sup>
- 1. School of Oceanography, Shanghai Jiao Tong University, Shanghai, China
- Corresponding author
- \*Correspondence to: Yuntao Zhou, ytzhou@sjtu.edu.cn, ORCID: 0000-0001-9714-5385

## 10 Abstract

Oceanic oxygen levels, crucial for marine ecosystems and biogeochemical cycles, have declined 12 significantly over the past few decades, driven by climate change and posing severe 13 environmental risks. However, historical dissolved oxygen (DO) measurements, especially below 14 2000 m, remain sparse, limiting comprehensive annual and seasonal analyses. Here we introduce 15 the BEM-DOR framework, a Bayesian-optimized ensemble of six machine-learning models 16 (Random Forest, XGBoost, LightGBM, CatBoost, Extremely Randomized Trees and 17 Histogram-based Gradient Boosting) fused via dynamic weighting, to reconstruct global monthly DO distributions at 1  $^\circ~$   $\times$  1  $^\circ~$  resolution from the surface to 5902 m depth over 1960-2023. 18 19 Validation against an independent dataset demonstrates that BEM-DOR outperforms existing 20 products. Our dataset captures depth-dependent deoxygenation, with the most pronounced 21 decline occurring between 150 and 200 m, and reveals dramatically accelerated oxygen loss in 22 the Arctic Ocean and North Atlantic over the past decade. We quantify uncertainties from 23 measurement errors, gridding processes, and model algorithms, providing the first long-term, 24 high-resolution, uncertainty-quantified DO product from ocean surface to nearly 6000 m depth. 25 The extension of DO data into the bathypelagic zone in this work is a significant contribution to 26 deep ocean oxygen dynamics and global biogeochemical cycles.

## 27 Keywords

dissolved oxygen, machine learning, ensemble learning, Bayesian optimization, data 29 reconstruction

## 31 1 Introduction

Over the past few decades, dissolved oxygen (DO) levels in open oceans have been continuously 33 decreasing (Breitburg et al., 2018; Keeling et al., 2010), primarily driven by climate change 34 (Deutsch et al., 2011). This decline has severe impacts on marine organisms and biogeochemical 35 processes, disrupting marine productivity, biodiversity, and biogeochemical cycles (Gruber, 2011; 36 Stramma et al., 2012). Climate models predict that global warming will further accelerate this 37 deoxygenation (Oschilies et al., 2018), potentially adversely affecting aerobic marine organisms 38 within this century (Sampaio et al., 2021), and altering biogeochemical cycles (Gruber, 2004; 39 Berman-Frank et al., 2008). Therefore, it is important to develop a comprehensive, 40 high-resolution reconstruction of ocean DO across both space and depth to accurately quantify 41 historical deoxygenation trends, identify regional hotspots, and inform future ecosystem and 42 climate projections.

Despite significant progress in oceanographic data collection, severe gaps in historical DO data 45 persist, hindering comprehensive analysis. For instance, the World Ocean Database (WOD) 46 (Mishonov et al., 2024) compiles DO profiles from research cruises and floats, yet most ocean 47 regions still lack any observations. This sparse spatial coverage severely This sparse spatial 48 coverage severely limits the use of data imputation methods to reconstruct planar or 49 three-dimensional DO fields. Furthermore, although many Earth System Models (ESMs) attempt 50 to simulate global oceanic DO, these models lack adjustments based on DO observation data, 51 leading to error propagation (Pathak et al., 2023). Thus, numerical models diverge significantly 52 from in-situ observations and consistently underestimate the actual DO decline trends (Bopp et 53 al., 2013; Cocco et al., 2013; Long et al., 2016; Kwiatkowski et al., 2020), restricting studies 54 related to ocean deoxygenation, Oxygen Minimum Zones (OMZs), biogeochemical cycles, and so 55 forth.

Classical geostatistical and interpolation methods have long been employed to map oceanic DO. 58 Zhou et al. (2022) combined geostatistical regression with Monte Carlo methods to estimate 59 changes in the area of Oxygen Minimum Zones (OMZs) globally and regionally from 1960 to 2019. 60 Garcia et al. (2024) applied objective analysis in WOA23 to produce internally consistent annual 61 and monthly DO fields from 1965 to 2022. Gouretski et al. (2024) developed an automated 62 quality control procedure to detect outliers and correct biases in ocean oxygen profiles, 63 producing a consistent global dataset from 1920 to 2023. Roach and Bindoff (2023) used Data 64 Interpolating Variational Analysis (DIVA) to generate a global high-resolution oxygen atlas from 65 1955 to 2018. Recently, machine learning methods exhibit higher computational efficiency, 66 capable of rapidly processing large-scale datasets. Giglio et al. (2018) utilized a Random Forest 67 Regression Model to present an estimate of oxygen at 150 m in the Southern Ocean based on 68 Argo data during 2008-2012. Sharp et al. (2022) reconstructed a global DO dataset called 69 GOBAI-O2 using feedforward neural networks and Random Forest Regression, spanning the years 70 2004 – 2022 with a monthly resolution, and extending from the ocean surface to a depth of 2 km. 71 Ito et al. (2024) developed a machine-learning ensemble of neural networks and random forests 72 trained on historical shipboard and biogeochemical Argo O2 profiles to generate gridded monthly 73 oxygen fields. While some of the DO data reconstruction studies focus on specific regions, some

span longer time spans, and some achieve higher temporal or spatial resolutions, it is challenging

- to simultaneously address all aspects.

Here we introduce the Bayesian Ensemble Machine-learning Dissolved Oxygen Reconstruction 78 (BEM-DOR) framework , which integrates six tree-based learners, Random Forest, XGBoost, 79 LightGBM, CatBoost, Extremely Randomized Trees and Histogram-based Gradient Boosting, each 80 tuned via Bayesian hyperparameter optimization. Model outputs are fused with dynamic "soft" 81 weights combining global cross-validation skill and local error performance. BEM-DOR produces a 82 global 1  $^{\circ}$   $\times$  1  $^{\circ}$ monthly DO dataset from 1960 to 2023 down to 5902 m, filling critical 83 deep-ocean gaps. We validate against The Global Ocean Data Analysis Project version 2 84 (GLODAPv2) (Olsen et al., 2016) with eight-fold temporal cross-validation, compare spatially with 85 Gridded Ocean Biogeochemistry from Artificial Intelligence - Oxygen (GOBAI) (Sharp et al., 2022), 86 ITO's product (Ito et al., 2024) and World Ocean Atlas 2023 (WOA23) (Garcia et al., 2024), and 87 quantify measurement, gridding and algorithm uncertainties. Finally, we analyze global, basin and 88 depth-resolved DO distribution and deoxygenation trends. We divided the global ocean into ten 89 basins to capture regional differences in oxygen storage and trends: North Pacific (NP), Equatorial 90 Pacific (EP), South Pacific (SP), North Atlantic (NA), Equatorial Atlantic (EA), South Atlantic (SA), 91 Indian Ocean north of the equator (NI), Indian Ocean south of the equator (SI), Southern Ocean 92 (SO) and Arctic Ocean (AO). Basin boundaries follow Schmidtko et al. (2017). 93

## 94 **2 Data and methods**

2.1 Data

#### 96 2.1.1 In-situ data of dissolved oxygen

We assembled our observational DO database by merging quality - controlled profiles from the

Array for Real-Time Geostrophic Oceanography dataset (Argo, https://argo.ucsd.edu) (Wong et al.,

2020) with CTD and OSD measurements archived in the World Ocean Database (WOD,

- https://www.ncei.noaa.gov/products/world-ocean-database) (Mishonov et al., 2024). Each
- profile consists of oxygen concentrations sampled at multiple depths at a given date and location.

We retained only those records flagged as good, then de - duplicated overlapping casts by

keeping the version with finer vertical sampling. We discarded any profile showing unrealistically

high or low values and excluded casts in which oxygen fell below 10  $\,\mu$  mol kg<sup>-1</sup> at any depth,

which likely indicate incorrect unit descriptions. Following Schmidtko et al. (2017), we treated the

combined dataset as free of systematic errors.

## 107 2.1.2 Reanalysis data of environmental factors

In this study, we investigated the contribution of physical, chemical, and biological factors to ocean DO during the period of 1960 –2023. We obtained monthly ocean temperature (T,  $^{\circ}$ C), salinity (S, ‰), meridional velocity and zonal velocity (MV and ZV, m/s) from the Ocean Reanalysis System 5 (ORAS5) gridded ocean dataset with a spatial resolution of 0.25°×0.25° and 75 vertical levels (Table S2), ranging from ocean surface to nearly 6000 m in depth (https://cds.climate.copernicus.eu/datasets/reanalysis-oras5).

#### 115 2.2 BEM-DOR framework

We developed the BEM-DOR (Bayesian Ensemble Machine-learning Dissolved Oxygen 117 Reconstruction) framework to reconstruct a global, monthly DO product from 1960 through 2023. 118 The process begins with assembling and preprocessing all available in situ DO profiles alongside 119 key environmental factors, such as temperature, salinity, and currents, onto a monthly grid 120 featuring 1°×1° horizontal resolution and 75 vertical levels. Next, each of six tree-based learners 121 (Random Forest, XGBoost, LightGBM, CatBoost, Extremely Randomized Trees and 122 Histogram-based Gradient Boosting) undergoes Bayesian hyperparameter tuning via Optuna's 123 TPE sampler, ensuring that each model's configuration minimizes cross-validation RMSE (Akiba et 124 al., 2019). Once optimized, the models train on the full gridded dataset and predict DO at every 125 valid grid cell, producing six complete five-dimensional DO fields. Those outputs are then merged 126 through a dynamic weighting scheme: global "prior" weights reflect each model's time-CV skill, 127 while local "dynamic" weights adjust according to the magnitude of agreement with nearby 128 observations, yielding a soft-weighted ensemble that adapts in space and time (Dietterich, 2000). 129 Finally, we validate and quantify uncertainty by performing eight-fold temporal cross-validation, independent evaluation against GLODAPv2 (Olsen et al., 2016) and comparisons with GOBAI 130 131 (Sharp et al., 2022), ITO (Ito et al., 2024) and WOA23 (Garcia et al., 2024). We then analyze depth 132 and basin-resolved deoxygenation trends to reveal the full vertical and regional patterns of 133 oxygen change.

#### 134 2.2.1 Data processing

In this study, all ocean DO observation data include temporal and spatial information, including 136 year, month, day, longitude, latitude, and measurement depth. Longitude and month are both 137 periodic features. For instance, longitude ranges from  $0^{\circ}$  to  $360^{\circ}$ , with  $360^{\circ}$  overlapping 138 with  $0^{\circ}$ , and months cycle annually. To address this issue, we followed the approach of Gade 139 (2010) and Tang et al. (2019) by converting the longitude and month attributes to polar 140 coordinate systems, using sine and cosine functions to simulate these features, thus preserving 141 their cyclical nature in the model.

$$coordinates = \begin{pmatrix} sin(latitude \cdot \frac{\pi}{180}) \\ sin(longitude \cdot \frac{\pi}{180}) \cdot cos(latitude \cdot \frac{\pi}{180}) \\ - cos(longitude \cdot \frac{\pi}{180}) \cdot cos(latitude \cdot \frac{\pi}{180}) \end{pmatrix}$$
 (1)

$time = \begin{pmatrix} \cos(month \cdot \frac{2\pi}{12})\\ \sin(month \cdot \frac{2\pi}{12}) \end{pmatrix}$

We use the ORAS5 reanalysis grid  $(1^{\circ} \times 1^{\circ})$  horizontal, 75 depth levels) as the target for 146 gridding all variables. DO observations are binned to each grid cell by averaging all points that fall 147 within the cell the same month and depth level. We upscaled the other environmental factors 148 with finer resolution, using inverse distance weighting of surrounding pixel values to match these 149 resolutions. To address potential multicollinearity, which can lead to instability in subsequent 150 modeling and increase the risk of overfitting, we analyzed correlations between the 11 factors.

(2)

- No correlation coefficient exceeded 0.4, so variable selection was not necessary in this case. A
- complete list of predictors, with abbreviations and data sources are shown in Table 1.

| Predictor                                                                            | Abbreviation | Product/Reference    |
|--------------------------------------------------------------------------------------|--------------|----------------------|
| $sin(latitude \cdot \pi/180)$                                                        | coord_1      | WOD (CTD+OSD) & Argo |
| $\sin(\operatorname{longitude} \pi/180) \cdot \cos(\operatorname{latitude} \pi/180)$ | coord_2      |                      |
| -cos(longitude $\pi/180$ ) ·cos(latitude $\pi/180$ )                                 | coord_3      |                      |
| Year                                                                                 | Year         |                      |
| $\cos(\mathrm{month} \cdot 2\pi/12)$                                                 | time_cos     |                      |
| $\sin(\mathrm{month}\cdot 2\pi/12)$                                                  | time_sin     |                      |
| Depth                                                                                | Depth        |                      |
| Temperature                                                                          | Т            | ORA-S5               |
| Salinity                                                                             | S            |                      |
| Zonal Velocity                                                                       | ZV           |                      |
| Meridional Velocity                                                                  | MV           |                      |

Note: The observational data come from WOD and Argo. The data from ORA-S5 are 0.25°x0.25°

monthly mean profile data.

#### 157 2.2.2 Machine learning models

We used six tree- based algorithms to reconstruct dissolved oxygen. Each model offers a different 159 balance of bias, variance and speed. We chose them for their strong performance in regression 160 tasks and their ability to handle nonlinear relationships. All models were trained on the same 161 input features and tuned via Bayesian optimization (Sect. 2.2.3). Below we describe each model. 162 Random Forest (RF) builds many decision trees on bootstrap samples and averages their outputs 163 (Breiman, 2001). It selects a random subset of features at each split. This randomness reduces 164 overfitting. RF handles large datasets well and is robust to outliers. CatBoost is a 165 gradient-boosting library designed for categorical features (Prokhorenkova et al., 2018). It uses 166 ordered target statistics to avoid target leakage. It grows symmetric trees and applies efficient 167 leaf pruning. CatBoost often converges faster and needs less tuning of learning rate. XGBoost 168 implements gradient boosting with second-order optimization (Chen & Guestrin, 2016). It adds 169 regularization to control tree complexity. It uses approximate split finding to speed up training on 170 large data. XGBoost balances accuracy and runtime efficiency. LightGBM uses histogram-based 171 binning and leaf-wise tree growth (Ke et al., 2017). It buckets continuous features into bins, 172 reducing memory. Trees grow by selecting splits that yield the largest loss reduction. LightGBM is 173 highly efficient for large feature sets and large datasets. Histogram-based Gradient Boosting 174 (Hist GBT) follows Friedman's original gradient boosting framework (Guryanov, 2019; 175 Friedman, 2001). It fits a sequence of weak learners to the negative gradient of the loss. It also 176 uses histogram binning for faster split evaluation. Hist GBT offers good accuracy in 177 high-dimensional settings. Extremely Randomized Trees (ERT) introduces extra randomness 178 compared to RF (Geurts et al., 2006). It picks split thresholds at random rather than searching for 179 the best cut. It uses the full dataset, rather than bootstrapping. This strong randomization further

180 lowers variance at modest cost in bias.

#### 181 2.2.3 Bayesian parameter optimization

To optimize hyperparameters across different machine learning models in a systematic and
efficient manner, we employed Bayesian optimization using the Optuna framework (Akiba et al.,
2019). This approach selects hyperparameter configurations based on the history of performance
evaluations, aiming to minimize the prediction error of each model.
Bayesian optimization constructs a probabilistic surrogate model of the objective function f(x),
where x is a vector of hyperparameters. The optimization seeks to identify the optimal x\* that
minimizes f:

$$\mathbf{x}^* = \arg\min_{x \in \chi} f(x) \tag{3}$$

- Here,  $\chi$  denotes the hyperparameter space. Optuna models the objective function using a
- Tree-structured Parzen Estimator (TPE), which fits two probability densities: one for good
- parameter configurations and another for all others. The next sampling point is chosen tomaximize the Expected Improvement (EI):

$$EI(X) = \int_{-\infty}^{y^*} (y^* - y) \cdot p(y \mid x) dy$$
 (4)

- where y\* is the current best objective value. The sampling focuses on regions with high EI.
- To reduce temporal overfitting and preserve model generalizability across decades,
- hyperparameter optimization was conducted using a subset of data from eight years (1960, 1968,
- 1976, 1984, 1992, 2000, 2008, 2016). The objective function minimized the Mean Squared Error
- (MSE) on an independent validation set derived from other eight test years (1967, 1975, 1983,
- 1991, 1999, 2007, 2015, 2023). The objective function was defined as:

$$MSE = \frac{1}{n} \sum_{i=1}^{n} (\hat{y}_i - y_i)^2$$
 (5)

- Each model was optimized over its own hyperparameter space, with the best-performing
  configuration recorded for final training and subsequent prediction on independent test data.
  This consistent, data-driven approach ensured fair comparability across all six learners and
  minimized bias from manual tuning. Below we summarize the search space and optimal
  parameters in Table 2.

| Model    | Hyperparameter      | Search Range          | Best Value            |
|----------|---------------------|-----------------------|-----------------------|
| ERT      | n_estimators        | 50 - 500              | 482                   |
|          | max_depth           | 3 - 20                | 20                    |
|          | min_samples_split   | 2 - 20                | 6                     |
|          | min_samples_leaf    | 1 - 10                | 2                     |
|          | max_features        | 0.1 - 1.0             | 0.856                 |
|          | bootstrap           | {True, False}         | False                 |
| CatBoost | iterations          | 50 - 1000             | 644                   |
|          | depth               | 3 - 12                | 7                     |
|          | learning_rate       | 0.005 - 0.3           | 0.246                 |
|          | l2_leaf_reg         | 10-5 - 10             | 7.18                  |
|          | random_strength     | 10-5 - 10             | 2.96                  |
|          | bagging_temperature | 0 - 1                 | 0.071                 |
|          | border_count        | 32 - 255              | 168                   |
| Hist_GBT | learning_rate       | 0.005 - 0.3           | 0.219                 |
|          | max_iter            | 50 - 1000             | 748                   |
|          | max_depth           | 3 - 12                | 9                     |
|          | min_samples_leaf    | 5 - 50                | 8                     |
|          | 12_regularization   | 10 <sup>-5</sup> - 10 | 1.24×10 <sup>-5</sup> |
|          | max_bins            | 32 - 255              | 187                   |
| LightGBM | n_estimators        | 50 - 1000             | 928                   |
|          | max_depth           | 3 - 12                | 9                     |
|          | learning_rate       | 0.005 - 0.3           | 0.132                 |
|          | num_leaves          | 10 - 300              | 118                   |
|          | min_child_samples   | 5 - 50                | 7                     |
|          | subsample           | 0.5 - 1.0             | 0.778                 |
|          | colsample_bytree    | 0.5 - 1.0             | 0.965                 |
|          | reg_alpha           | 10-8 - 10             | 3.24×10 <sup>-8</sup> |
|          | reg_lambda          | 10-8 - 10             | 0.187                 |
| RF       | num_trees           | 10 - 200              | 59                    |
|          | min_leaf_size       | 10 - 100              | 10                    |
| XGBoost  | n_estimators        | 50 - 1000             | 830                   |
|          | max_depth           | 3 - 12                | 12                    |
|          | learning_rate       | 0.005 - 0.3           | 0.265                 |
|          | min_child_weight    | 1 - 10                | 1                     |
|          | subsample           | 0.5 - 1.0             | 0.615                 |
|          | colsample bytree    | 0.5 - 1.0             | 0.908                 |

### 210 Table 2. Hyperparameter Search Spaces and Optimal Values

#### 212 2.2.4 Multi-model fusion and dynamic weighting strategy

We fuse six model predictions into one field. Our goal is to combine global model skill with local

fit to observations. We assign each model a static "prior" weight. We then adjust those weights

at each grid cell using the local agreement between prediction and observation.

 $216 \qquad \text{We derive a prior weight } w_i \text{ for model i from its time-cross-validation (Sect. 3.1) RMSE } \epsilon_i. \text{ We set a}$ 

decay parameter β. Then:

$$\omega_i = \frac{\exp(-\beta\varepsilon_i)}{\sum_{j=1}^{6} \exp(-\beta\varepsilon_j)}, \sum_{i=1}^{6} \omega_i = 1$$
(6)

- A smaller  $\epsilon_i$  yields a larger w<sub>i</sub>. We choose  $\beta$ =1 to balance influence among models.
- At each grid cell x, we compute a dynamic weight  $v_i(x)$ . We use a tuning parameter  $\alpha$ . For cells
- where an observation O(x) exists, we set:

$$v_i(x) = \exp(-\alpha | p_i(x) - O(x) |)$$
 (7)

Here  $p_i(x)$  is model i's prediction. A smaller local error makes  $v_i(x)$  larger. We use  $\alpha=1$ . For cells with no observation, we fall back on the static weight:

$$225 \quad \mathbf{v}_i(\mathbf{x}) = \boldsymbol{\omega}_i \tag{8}$$

We compute the ensemble prediction E(x) by normalizing the dynamic weights:

$$E(x) = \frac{\sum_{i=1}^{6} v_i(x) p_i(x)}{\sum_{i=1}^{6} v_i(x)}, if \sum_i v_i(x) > 0$$
(9)

and E(x)=NaN if all p<sub>i</sub>(x) are missing. This formula guarantees that models aligning well with local observations gain more influence, while the static weights keep poorly observed regions stable.

#### 230 **2.2.5 Data reconstruction**

We produce a global, monthly dissolved oxygen (DO) dataset on a regular  $1^{\circ} \times 1^{\circ}$  grid and 75 232 depth levels (0-5902 m) spanning 1960-2023. First, we gather all predictor fields described in 233 Table 1. Each field is remapped to the target grid and monthly time step following ORAS5. Next, 234 we apply the six optimized machine-learning models (Sect. 2.2.2) at every valid grid cell and time. 235 Each model ingests the full vector of predictors and returns a DO estimate only where all 236 predictors are present. This yields six parallel prediction arrays of dimensions 237  $360 \times 180 \times 75 \times 12 \times 64$ . We then merge these arrays using our dynamic weighting scheme 238 (Sect. 2.2.4). Static "prior" weights reflect each model's cross-validation skill. Local weights adapt 239 to agreement with any overlapping in situ observation. The weighted combination produces a 240 single ensemble DO field at each grid cell and month. We packaged the ensemble field into a 241 CF-compliant NetCDF file with coordinate variables, depth layers, time and global attributes 242 documenting methods.

### 244 **3 Model performance**

#### 245 3.1 Model Temporal Cross- Validation

We conducted eight-fold temporal cross-validation on each of the six models. In each fold f, data

- from eight test years  $\{1960 + f + 8k\}_{k=0}^{7}$  formed the test set, with remaining years for training.
- We trained each model using its optimized hyperparameters (Sect. 2.2.3) on the training years, 249 predicted the test years, and computed mean bias (ΔDO), mean absolute error (MAE), 250 root-mean-square error (RMSE) and coefficient of determination (R<sup>2</sup>) on the held-out data. These 251 metrics collectively provide a comprehensive understanding of the model's predictive accuracy

and bias. The results appear in Tables 3-6.

$$RMSE = \sqrt{\frac{1}{n} \sum_{i=1}^{n} (y_i - \hat{y}_i)^2}$$
 (10)

$$MAE = \frac{1}{n} \sum_{i=1}^{n} |y_i - \hat{y}_i|$$
 (11)

$$R^{2} = 1 - \frac{\sum_{i=1}^{n} (y_{i} - \hat{y}_{i})^{2}}{\sum_{i=1}^{n} (y_{i} - \overline{y})^{2}}$$
(12)

$$\Delta DO = \frac{1}{n} \sum_{i=1}^{n} y_i - \hat{y}_i$$
(13)

### 257 Table 2. Cross-validation ΔDO (µmol kg<sup>-1</sup>)

| Model                                                  | Fold 1 | Fold 2 | Fold 3 | Fold 4 | Fold 5 | Fold 6 | Fold 7 | Fold 8 |
|--------------------------------------------------------|--------|--------|--------|--------|--------|--------|--------|--------|
| RF                                                     | -0.160 | 0.253  | -0.089 | 0.134  | 0.085  | 0.155  | 0.444  | 0.347  |
| XGBoost                                                | -0.403 | 0.313  | -0.125 | 0.291  | -0.007 | 0.158  | 0.325  | 0.211  |
| LightGBM                                               | -0.470 | 0.289  | -0.173 | 0.250  | 0.048  | 0.193  | 0.391  | 0.255  |
| Hist_GBT                                               | -0.324 | 0.204  | -0.170 | 0.174  | 0.052  | 0.186  | 0.378  | 0.285  |
| ERT                                                    | -0.298 | -0.120 | -0.346 | -0.069 | -0.127 | 0.027  | 0.662  | 0.235  |
| CatBoost                                               | -0.287 | 0.128  | -0.134 | 0.244  | 0.081  | 0.072  | 0.505  | 0.165  |
| Table 3. Cross-validation MAE (μmol kg <sup>-1</sup> ) |        |        |        |        |        |        |        |        |
| Model                                                  | Fold 1 | Fold 2 | Fold 3 | Fold 4 | Fold 5 | Fold 6 | Fold 7 | Fold 8 |

|          |        |        |        |        |        |        | /      | •      |
|----------|--------|--------|--------|--------|--------|--------|--------|--------|
| RF       | 10.404 | 9.894  | 9.786  | 9.661  | 9.540  | 9.457  | 9.520  | 9.763  |
| XGBoost  | 10.943 | 10.471 | 10.406 | 10.281 | 10.130 | 10.011 | 10.130 | 10.320 |
| LightGBM | 10.849 | 10.376 | 10.328 | 10.136 | 10.011 | 9.921  | 9.964  | 10.218 |
| Hist_GBT | 11.510 | 11.045 | 10.926 | 10.799 | 10.671 | 10.588 | 10.663 | 10.926 |
| ERT      | 10.627 | 10.180 | 9.939  | 9.834  | 9.752  | 9.668  | 9.726  | 9.995  |
| CatBoost | 11.904 | 11.538 | 11.247 | 11.158 | 11.053 | 10.942 | 10.990 | 11.299 |

## 259 Table 4. Cross-validation RMSE (µmol kg<sup>-1</sup>)

| Model    | Fold 1 | Fold 2 | Fold 3 | Fold 4 | Fold 5 | Fold 6 | Fold 7 | Fold 8 |
|----------|--------|--------|--------|--------|--------|--------|--------|--------|
| RF       | 17.294 | 16.610 | 16.317 | 16.337 | 15.999 | 16.028 | 16.344 | 16.302 |
| XGBoost  | 17.620 | 16.968 | 16.814 | 16.829 | 16.479 | 16.391 | 16.756 | 16.752 |
| LightGBM | 17.405 | 16.789 | 16.634 | 16.506 | 16.260 | 16.273 | 16.526 | 16.503 |
| Hist_GBT | 18.048 | 17.477 | 17.130 | 17.154 | 16.892 | 16.859 | 17.210 | 17.179 |
| ERT      | 17.325 | 16.801 | 16.238 | 16.433 | 16.016 | 16.094 | 16.321 | 16.349 |
| CatBoost | 18.501 | 18.068 | 17.464 | 17.607 | 17.312 | 17.324 | 17.568 | 17.582 |

#### 260 Table 5. Cross-validation R<sup>2</sup>

| Model   | Fold 1 | Fold 2 | Fold 3 | Fold 4 | Fold 5 | Fold 6 | Fold 7 | Fold 8 |
|---------|--------|--------|--------|--------|--------|--------|--------|--------|
| RF      | 0.958  | 0.961  | 0.960  | 0.959  | 0.960  | 0.960  | 0.960  | 0.963  |
| XGBoost | 0.957  | 0.959  | 0.958  | 0.957  | 0.957  | 0.958  | 0.958  | 0.961  |

| LightGBM | 0.958 | 0.960 | 0.958 | 0.959 | 0.959 | 0.958 | 0.959 | 0.962 |
|----------|-------|-------|-------|-------|-------|-------|-------|-------|
| Hist_GBT | 0.954 | 0.957 | 0.956 | 0.955 | 0.955 | 0.955 | 0.956 | 0.959 |
| ERT      | 0.958 | 0.960 | 0.960 | 0.959 | 0.959 | 0.959 | 0.960 | 0.963 |
| CatBoost | 0.952 | 0.954 | 0.954 | 0.953 | 0.953 | 0.953 | 0.954 | 0.957 |

All six learners exhibit remarkably consistent skill across the eight temporal folds, with only minor 262 263 spread in error metrics (Table 2-5). LightGBM's MAE varies by less than 0.9 μmol kg<sup>-1</sup> (9.92–10.85) and its RMSE by under 0.9  $\mu$ mol kg<sup>-1</sup> (16.26–17.41), yielding an R<sup>2</sup> range of 0.958–0.962-small 264 fluctuations that underscore stable performance year-to-year. RF delivers the lowest RMSE 265 266 (15.99–17.29) and highest R<sup>2</sup> (0.958–0.963). In contrast, CatBoost and Hist\_GBT register higher mean errors (MAE up to 11.90 and 11.51, RMSE up to 18.50 and 18.05) and slightly larger 267 268 inter-fold variability, indicating they are more sensitive to the specific training/test split (Table 269 2–5). ERT and XGBoost fall between these extremes, with moderate error levels and consistent R<sup>2</sup>. 270 Crucially, no model ever produces an outlier fold with dramatically degraded skill-each 271 maintains MAE < 12  $\mu$ mol kg<sup>-1</sup> and R<sup>2</sup> > 0.95. This uniformity across folds confirms strong 272 temporal generalization and validates our choice of an ensemble approach (Bergmeir & Benítez, 273 2012; Roberts et al., 2017).

### 275 **3.2 Evaluation on independent observations**

We evaluated both the ensemble and each single model against an independent GLODAPv2 277 dissolved oxygen dataset, treated here as ground truth. GLODAPv2 profiles were averaged into 278 the same  $1^{\circ}\times1^{\circ}$  grid and monthly time step as the reconstructions. We then identified grid cells 279 where both the gridded GLODAPv2 values and model predictions were non-NaN. At those 280 collocated points we computed four summary metrics: mean bias ( $\Delta$ DO), mean absolute error 281 (MAE), root mean square error (RMSE) and coefficient of determination (R<sup>2</sup>).

#### 282 Table 6. Comparison of Ensemble and Single Models on GLODAPv2 Dataset

| Model           | MAE (µmol kg <sup>-1</sup> ) | RMSE (µmol         | R <sup>2</sup> | $\Delta { m DO}$ (µmol |
|-----------------|------------------------------|--------------------|----------------|------------------------|
|                 |                              | kg <sup>-1</sup> ) |                | kg <sup>−1</sup> )     |
| Ensemble        | 5.003                        | 9.895              | 0.985          | 0.311                  |
| Ensemble(static | 5.061                        | 9.982              | 0.985          | 0.341                  |
| weight=1)       |                              |                    |                |                        |
| RF              | 5.802                        | 10.979             | 0.982          | 0.221                  |
| XGBoost         | 6.198                        | 10.943             | 0.982          | 0.330                  |
| ERT             | 6.757                        | 11.960             | 0.979          | 0.405                  |
| LightGBM        | 7.637                        | 12.825             | 0.975          | 0.543                  |
| Hist_GBT        | 8.632                        | 13.875             | 0.971          | 0.589                  |
| CatBoost        | 9.179                        | 14.474             | 0.969          | 0.693                  |

<sup>283</sup> 

Table 6 also includes an equal-weight ensemble (static weight = 1), which yields MAE = 5.061  $\mu$ mol kg<sup>-1</sup>, RMSE = 9.982  $\mu$ mol kg<sup>-1</sup> and R<sup>2</sup> = 0.985 ( $\Delta$ DO = 0.341). Although this uniform blend already outperforms any single model, our RMSE-based prior weights push performance further, dropping MAE to 5.003  $\mu$ mol kg<sup>-1</sup>, RMSE to 9.895  $\mu$ mol kg<sup>-1</sup> and raising R<sup>2</sup> to 0.985, demonstrating that leveraging each learner's cross-validation skill yields a measurably better

ensemble than equal weighting. Among the individual algorithms, Random Forest (MAE 5.802,
RMSE 10.979, R<sup>2</sup> 0.982) and XGBoost (MAE 6.198, RMSE 10.943, R<sup>2</sup> 0.982) follow most closely,
while CatBoost (MAE 9.179, RMSE 14.474, R<sup>2</sup> 0.969) and Hist\_GBT (MAE 8.632, RMSE 13.875, R<sup>2</sup>
0.971) sit at the lower end. All models keep bias under 0.7 μmol kg<sup>-1</sup>. No single method ever
exhibits a catastrophic fold, underscoring the robustness of our dynamic weighting in combining
complementary strengths and minimizing weaknesses (Dietterich, 2000).

### 296 **3.3 Uncertainty estimations**

We quantify three independent sources of error in the reconstructed DO field. Estimating each 297 298 component lets us understand their relative contributions and report a rigorous total uncertainty. 299 Measurement uncertainty ( $\triangle O_{meas}$ ) originates from the inherent precision limits of the in situ 300 dissolved oxygen observations. In our study, we assume a constant uncertainty based on Ito et al., 301 (2024): OSD and CTD data are assigned a measurement uncertainty of 1 µmol kg<sup>-1</sup>, while Argo 302 data are attributed a value of 3  $\mu mol~kg^{-1}$ . Thus, for any given observation, we represent the 303 uncertainty as  $\triangle O_{meas}=1 \ \mu mol \ kg^{-1}$  for OSD or CTD data and  $\triangle O_{meas}=3 \ \mu mol \ kg^{-1}$  for Argo data. 304 This constant assignment provides a pragmatic baseline for quantifying the observational error in 305 the reconstructed dataset, acknowledging that, although regional variations might introduce 306 additional variability, such effects are not considered in this baseline estimate.

Grid uncertainty ( $\triangle O_{grid}$ ) quantifies the representation error associated with assigning a single 308 dissolved oxygen value to a 1°×1° spatial cell across time. To estimate grid uncertainty, 309 observations within each grid cell are collocated, and the standard deviation ( $\sigma$ ) among these 310 observations is computed. For a given grid cell, let the n observations be denoted by O<sub>1</sub>, O<sub>2</sub>, ..., 311 O<sub>n</sub>; then, the grid uncertainty is estimated by:

$$\Delta O_{\text{grid}} = \sigma = \sqrt{\frac{1}{n-1} \sum_{i=1}^{n} (O_i - \overline{O})^2}$$
 (14)

where  $\ O$  is the mean value of the observations in that cell. These standard deviations are then 313 314 averaged across the entire dataset, providing an overall estimate of the grid uncertainty. This 315 method effectively captures the dispersion due to variable sampling density and spatial 316 heterogeneity, reflecting the error introduced by mapping sparse in situ data onto a coarser grid. 317 Algorithm uncertainty (  $\triangle$   $O_{alg}$ ) reflects the error introduced by the machine learning 318 reconstruction process. In this study, six ensemble models (RF, XGBoost, LightGBM, CatBoost, ERT, 319 and Hist\_GBT) were trained using a comprehensive set of dissolved oxygen observations and 320 environmental factors. Each model was optimized via Bayesian hyperparameter tuning and 321 validated using an eight-fold cross-validation procedure, yielding an MAE for each model, 322 denoted by error<sub>1</sub> through error<sub>6</sub>. We then compute the prior weight for the i-th model using an 323 exponential decay function:

$$\omega_{i} = \frac{\exp(-error_{i})}{\sum_{j=1}^{6} \exp(-error_{j})}$$
(15)

The overall algorithm uncertainty is then estimated as the weighted average of the MAE values:

$$326 \qquad \Delta O_{alg} = \sum_{i=1}^{6} \omega_i error_i \tag{16}$$

This approach synthesizes the performance of the six different models into a single metric, 328 representing the inherent uncertainty of the reconstruction algorithm across the entire dataset. 329 The resulting component uncertainties are measurement uncertainty = 1.60  $\mu$ mol kg<sup>-1</sup>, grid 330 uncertainty = 4.61  $\mu$ mol kg<sup>-1</sup>, and algorithm uncertainty = 10.17  $\mu$ mol kg<sup>-1</sup>. Finally, the total 331 uncertainty in the reconstructed dissolved oxygen field is expressed as

$$\Delta O_{\text{total}} = \sqrt{\Delta O_{meas}^2 + \Delta O_{grid}^2 + \Delta_{alg}^2} = 11.28 \,\mu\text{mol kg}^{-1}$$

 $333 \qquad \text{This formulation integrates the observational, mapping, and model reconstruction uncertainties}$ 

- into a comprehensive framework for error quantification.

## **4 Data product comparison**

## 337 4.1 Comparison with GLODAPv2 observations

To rigorously evaluate our DO reconstruction, we conducted a systematic comparison with two 339 recent datasets, GOBAI (Sharp et al., 2023) and ITO (Ito et al., 2024), using the quality-controlled 340 GLODAPv2 dataset (Olsen et al., 2016) an independent validation standard. Our reconstruction 341 consistently achieves the lowest MAE and RMSE, and the highest R<sup>2</sup>, with near-zero bias. Within 342 the spatial, temporal, and depth ranges of GOBAI, our model also outperforms GOBAI by 15-20% 343 in both MAE and RMSE, and by 0.01 in R<sup>2</sup>. Within the spatial, temporal, and depth ranges of ITO, 344 our model show a 30% reduction in MAE and RMSE compared to ITO, while the bias improved 345 significantly, decreasing from 1.74 to 0.34  $\mu mol~kg^{\text{-1}}.$  These comparisons confirm that our 346 ensemble delivers consistently better agreement with independent GLODAPv2 observations, 347 both globally and within the individual domains of GOBAI and ITO.

| Product        | MAE   | RMSE   | R <sup>2</sup> | $\Delta$ DO |
|----------------|-------|--------|----------------|-------------|
| Our            | 5.003 | 9.895  | 0.985          | 0.311       |
| reconstruction |       |        |                |             |
| (full          |       |        |                |             |
| GLODAPv2)      |       |        |                |             |
| GOBAI on       | 8.107 | 14.201 | 0.969          | 1.167       |
| GLODAPv2       |       |        |                |             |
| Our            | 6.399 | 12.353 | 0.980          | 1.019       |
| reconstruction | in    |        |                |             |
| GOBAI          |       |        |                |             |
| coverage       |       |        |                |             |
| ITO on         | 9.417 | 16.790 | 0.961          | 1.742       |
| GLODAPv2       |       |        |                |             |
| Our            | 6.507 | 11.991 | 0.981          | 0.344       |
| reconstruction | in    |        |                |             |
| ITO coverage   |       |        |                |             |

#### 348 Table 7. Performance comparison on GLODAPv2