# Peer review of "Reconstructing Global Monthly Ocean Dissolved Oxygen"

_Earth System Science Data, 2025_

## Referee Comment (RC1)

**General comments**

I have read and reviewed the manuscript titled "*Reconstructing Global Monthly Ocean Dissolved Oxygen (1960–2023) to Nearly 6000 m Depth Using Bayesian Ensemble Machine Learning*" by Mingyu Han and Yuntao Zhou. In the article, the authors present a new data product (BEM-DOR) of monthly dissolved oxygen concentrations in the global ocean, from the surface to 5902 m depth, built using an ensemble of decision-tree machine learning models. The models are trained on a combined dataset of *in situ* dissolved oxygen observations from the World Ocean Database 2023 and Argo floats (target) with ORAS5 model output for oceanographic variables such as temperature, salinity, zonal velocity, meridional velocity and geospatial coordinates (features).

While the product represents an advancement compared to the data products already available, as it expands the vertical coverage of dissolved oxygen products built using machine learning, I have several concerns on the methodology and results presented in the paper. Moreover, the discussion of the new BEM-DOR product lacks deep contextualisation and comparison with the products already available.

Additionally, the assets available for review only include the final dataset in netcdf format. In line with good practices in the field of machine learning, I would like the authors to make their code available for reproducibility testing.

Overall, the text requires some substantial re-writing and the authors need to provide additional evidence to some of their claims. The language used to describe results and discuss the BEM-DOR data product's reconstructions compared to other available products is at times misleading and needs to be changed. Only after the authors have addressed my comments and feedback, I will be happy to reconsider this manuscript for publication.

**Specific comments**

L103-105: provide a quantitative definition of unrealistically high or low. Also, the arbitrary exclusion of casts where any reading is below 10 μmol/kg would exclude areas of severe hypoxia and low oxygenated waters. If the authors followed an established methodology, I would want to see a reference to it. Otherwise, I would suggest their method to be at least partially reconsidered.

L110: what is the rationale behind the inclusion of zonal and meridional velocities as environmental predictors for dissolved oxygen concentrations?

L145: I understand from L111 and the documentation of ORAS5 that the data is gridded at 0.25° x 0.25° resolution. Where is this 1° x 1° grid coming from?

L157: why did the authors decide to use six models in the ensemble? And why are all the algorithms tree-based? Please explain further in the text.

L164-165: why did the authors include CatBoost in the ensemble if there are no categorical features in the framework proposed (BEM-DOR)?

L210: how were the hyperparameters to be tuned chosen? And how was the search range identified / selected?

L228: where would all predictions be missing? On land? Or at locations where no observations are available in the validation split during cross-validation? Please clarify further.

L246-250: it is unclear to me how this temporal cross-validation differs from the cross-validation done for hyperparameter tuning. First, I would like to have a more detailed explanation of what years formed the test set and what the training set, as the expression provided in line 247 is not clear. Additionally, I would like to see a detailed clarification of the differences between hyperparameter-tuning cross-validation and temporal cross-validation and the rationale behind cross-validating twice in model development.

L275-294 (Sect. 3.2) and then 337-366 (Sect. 4.1): Did the authors make sure that the observations they validate against in GLODAPv2 are not also included in the World Ocean Database 2023? Otherwise, they might validate against the same observations they are using to train the model. Similarly, the GOBAI-$O_2$ product (Sharp et al., 2023) is built using GLODAPv2 observations as training data, and the product of Ito et al. (2024) is built on World Ocean Database 2018 data. How did the authors ensure that their validation data were not included in the training of these two models as well?

L338: could the authors please provide a detailed description of how the comparison was performed, as it is unclear in the text?

L368: why do the authors not include the product of Roach & Bindoff (2023) in their comparison in Sects. 4.2 and 4.3, especially as that product is available up to depths of 6800 m?

L377-384: the exact difference between the lines is hard to quantify from the graph, but the authors claim that the difference between their work / WOA2023 and Ito is 2-5 umol/kg between 800-1000m when the lines seem to overlap. At the same time, they say that the difference between their study and WOA23 is 2-3 umol/kg at deeper depths down to 5902 m, while the graph clearly shows the lines diverging. This paragraph needs to be revised and the discrepancy in analytical interpretation addressed.

L417-429 (Sect. 5.1): this section does not add much to what is already known from a scientific perspective about large scale dissolved oxygen distribution. I suggest the authors delve deeper into some specific features of the data product that are novel compared to what is already available in the literature to provide additional evidence of why their data product is valuable.

L436-464 (Sect. 5.2): similarly to the section (and comment) above, Sect. 5.2 only provides rather general and already well-known descriptions of the variations of mean dissolved

oxygen concentrations throughout the water column. Additionally, the mean dissolved oxygen concentration profile in Fuigure 5 is the same as the one plotted in Figure 2.

Dataset in netcdf format: the values of 'time' and 'depth' seem to be decoded incorrectly in the final version of the file. Time is only reported as timesteps (0 to 767; without any decodable information on month or year). When opening the file in 'ncview', depth is only readable as depth level (1 to 74, without any information on the depth value in meters). Lastly, latitude, longitude and depth are included in the dataset as variables instead of coordinates.

**Technical corrections**

L47: repetition of 'This sparse spatial coverage severely'.

L52 and throughout the text: *in situ* should be written in italics and without a dash.

L53-55: what do the authors mean with 'restricting'? This sentence is phrased awkwardly and needs clarifying.

L60: define what WOA23 is.

L65-66: very vague and broad sentence. Provide references in the context of Earth sciences and oceanography.

L68 and throughout the text: the reference to the GOBAI-$O_2$ product is incorrect. It is Sharp et al. (2023): *Sharp, J. D., Fassbender, A. J., Carter, B. R., Johnson, G. C., Schultz, C., & Dunne, J. P. (2022). GOBAI-$O_2$: temporally and spatially resolved fields of ocean interior dissolved oxygen over nearly two decades. Earth System Science Data, 2023, 15, 10, 4481-4518.*

L69 and throughout the text: it is $O_2$ rather than O2.

L73-75: provide references.

L86: Ito et al. (2024) is missing in the reference list.

L99: define what OSD and CTD mean.

L106: how many observations are there in the final dataset? What is their distribution in space and time? The latter question can be answered by providing an additional figure either in the text or Supplementary Information.

L108: what biological factors? None of the environmental factors included in the models represent biology.

L110: salinity is expressed as PSU and it is unitless.

L112: be more specific. Is it 5902 m?

L115-133 (Sect. 2.2): I find this paragraph quite hard to follow for the average reader, as there are many undefined abbreviations and technical terms. I would suggest simplifying it and referring the reader to the more detailed subsections that follow in the text.

L125: what does 'producing six complete five-dimensional DO fields' mean? Please clarify.

L151: could the authors provide a table (in Supplementary Information) of the correlation values?

L153: there are only 11 environmental factors in the table, while the caption mentions 19. Correct typo.

L157-180 (Sect. 2.2.2): This paragraph needs strong rewording. As it is, it provides very general descriptions of the models using algorithm-specific terminology that might not be familiar to the readers. Moreover, the descriptions are very surface-level, and the authors do not provide any references for the claims they make regarding the different algorithms.

L184-185: what do the authors mean with 'history of performance evaluations'? Please clarify.

L190-192: Please rewrite this sentence with more details in order to make it more understandable to the reader.

L196-200: what is the size of the training and validation sets defined for cross-validation?

L218 and 222: ensure consistency between alphabets used. In equation 6, w is not $\omega$ and, in equation 7, α is not $\alpha$.

L237: name dimensions together with their sizes so the reader does not have to guess or do the math while reading.

L241: spell out what CF-compliant means.

L257: Table 2 should be labelled as table 3, and so on. There is already a table 2 at line 210. Additionally, I suggest clarifying further what years are included in each fold, either by providing a schematic representation or a more detailed explanation in the text. As they are, the tables are not very self-explanatory and do not add much to the results presented, so they could be moved to the Supplementary Information.

L293: 'catastrophic fold' is not scientific terminology. Please change accordingly.

L312: I would argue that for the readership of ESSD, it is unnecessary to include here the formula of standard deviation.

L397: correct typo. 'Light' should be 'slight'.

L398: based on plot A in Figure 3, differences at high latitudes seem much larger than ±2 μmol/kg. Please revise.

L409: change 'modest' with 'larger'.

L412: As for my comment above, the differences observed in plot G (Figure 3) seem higher than ±3 μmol/kg at some locations. Please revise.

L430: change subplot titles of Figure 4 to match the four depth levels mentioned in the text and caption.

L468: why is oxygen content plotted as a scatter plot when the values are completely independent of each other? Please change.

L502: this is a bold claim considering that the profiles nearly overlap in figure 2. Please tone it down.

Figures: all figures seem to be in low definition. Please provide higher-quality images.

---

## Referee Comment (RC2)

In "Reconstructing Global Monthly Ocean Dissolved Oxygen (1960-2023) to Nearly 6000 m Depth Using Bayesian Ensemble Machine Learning", the authors use an ensemble of six machine learning models to reconstruct 4D fields of oxygen going back to 1960. They train their models on a mix of Argo oxygen and shipboard data in the World Ocean Database, then create a composite reconstruction that tries to weight the final output toward the models that perform better in a given location and time.

This work is timely, as many groups are trying to make use of the recent increases in oxygen data to create temporally- and spatially-resolved oxygen maps. It is important, as oxygen is an essential ocean variable and can provide a significant amount of information on its own as well as provide useful context to a wide range of oceanographic studies. However, I have several concerns about important details of their work and conclusions regarding ocean deoxygenation rates. Unfortunately, given the apparent use of overlapping datasets for validation and for training, I do not think the manuscript is suitable for publication in its current state and the analysis would need to be redone entirely. I therefore recommend that this manuscript be rejected, though I do hope that it is resubmitted after these issues are dealt with.

Sincerely,

Seth Bushinsky, University of Hawai'i at Mānoa

**Main issues:**

World Ocean Database and GLODAP are not independent of one another. There is overlap between the datasets and it is not clear from the text that this was considered and the overlapping cruises removed. Given that no mention of this is made in the text, my guess is that the "independent" evaluation is actually just evaluating the oxygen product with a subset of the training data, in which case it is no surprise that your product performs so well. Please clarify if you did, in fact, remove GLODAP cruises from WOD, and if not, then I would rethink your approach to training and validation.

Another point is that much of this manuscript seems to be written for machine learning researchers to understand, not chemical oceanographers who use oxygen. To an extent, that makes sense, but it would be very beneficial if you could explain the choices made in the methods (especially 2.2.3 and 2.2.4) so that those who do not run ML models can understand their import and your rationale.

Uncertainty estimates: Currently you only are including in individual measurements (i.e. sensor precision) but ignoring the impact of potential biases, as we and others have

recently shown exist in the float oxygen dataset (Gouretski et al 2024, Bushinsky et al 2025, etc). You also assume that your observations within a given grid cell adequately capture the range of uncertainty, when this ignores the fact that a few samples within a 1x1 monthly grid cell are unlikely to capture the range in environmental variability. I am glad that you do include an uncertainty estimate, but with some adjustments it might do a better job of reflecting true uncertainty in your reconstructions.

Global deoxygenation rates: Your conclusion of dramatic increases in deoxygenation since 2010 need to be re-examined in light of biases in the Argo oxygen dataset. In addition to the ones mentioned above, optode-based oxygen sensors, which represent roughly half of the Argo oxygen data, have a relatively slow response time and are therefore known to be biased in the thermocline (typically toward low values) / regions of strong oxyclines. This may be part of the strong rate of change in oxygen seen in the blue line in Figure 5 between ~50m and 500m and described in the text. You cite Bittig's paper on this topic and mention it in your conclusions but need to consider the importance when evaluating the apparent decreases in oxygen content that you describe. Also, some assessment of the uncertainty in your rates should be included.

**Other comments:**

Line 47 – sentence fragment needs to be removed

Line 104 – why would you assume oxygen below 10umol/kg is an issue with unit descriptions? What about oxygen deficient/minimum zones? If there was an incorrect unit (i.e. should have been ml/l) that would surely affect the whole profile, not just an individual measurement. This filter does not make sense to me.

Line 125 – 5 dimensional fields? What is the fifth dimension?

Line 126 – what is CV?

Line 145-147 – How do you deal with the different vertical sampling resolutions of bottle and, CTD, and float profiles? If you are just averaging all points you would be weighting bottle measurements the least, which doesn't make sense as those are the highest accuracy data that you have.

Table 1 – title mentions 19 environmental factors, text says 11 predictors, but 3 seem to be time and 3 seem to be position, so I'm not sure whether either descriptor is correct.

Line 252- I think this should read Tables 2-5.

Figure 2. It is difficult to see the differences that you discuss in the text. I would recommend adding a subplot to the right that shows the differences relative to WOA as a function of depth.

Lines 395-413: It seems like in this paragraph you both say that it is good when you match WOA but then say that when you disagree it means that your product is better. Both cannot be true without more of a rationalization for that conclusion. Also, why assume that WOA is best? Maybe ITO is better when there are disagreements, or maybe your product is. Please include the rationale for trusting the WOA maps.

Data availability – no mention of what version of WOA, GLODAPv2 (what year?), WOD, etc. Need to be included.

---

## Author Comment (AC2)

Dear Referee,

We are grateful for the referee's thoughtful, detailed comments. We have prepared detailed responses to each comment and outlined the specific revisions that we will incorporate into the manuscript in the revised submission. The major updates include: **(i) a strict GLODAPv2-WOD23-Argo de-duplication to ensure an independent validation set, with figures and tables updated (Figs. R2-R5; Tables R1-R2); (ii) new Section 5.3 on hemispheric seasonal cycles, quantifying deep-ocean amplitudes to 5902 m; (iii) new Section 5.4 with latitude-depth transects at 160°W and 30°W showing means and multi-decadal trends; (iv) the addition of comparisons with the Roach & Bindoff product; (v) clarified rationale for including zonal/meridional velocities, the choice of six tree-based learners, and our two-stage cross-validation design; and (vi) an explicit explanation of the 10 µmol kg$^{-1}$ profile-level QC rule with an illustrative example.** A point-by-point reply follows below (referee comments in *italics*, our responses in regular type).

On behalf of all authors, sincerely,

Mingyu Han, Yuntao Zhou
Shanghai Jiao Tong University

**Comment 1:**

*L103-105: provide a quantitative definition of unrealistically high or low. Also, the arbitrary exclusion of casts where any reading is below 10 µmol/kg would exclude areas of severe hypoxia and low oxygenated waters. If the authors followed an established methodology, I would want to see a reference to it. Otherwise, I would suggest their method to be at least partially reconsidered.*

**Response 1:**

Thank you for pointing this out. We used an internal QC to set the threshold and applied it at the profile level: a profile was discarded only if dissolved oxygen was < 10 µmol kg$^{-1}$ at every sampled depth. Profiles that never rise above this level are overwhelmingly associated with sensor or unit-conversion problems (Fig. R1). In other words, the filter targeted profiles that were uniformly hypoxic from surface to bottom. With this clarification, the procedure does not remove legitimate observations from oxygen minimum zones such as the eastern tropical Pacific or Arabian Sea.

[Figure]

**Figure R1. Example of a profile removed by the profile-level low-O$_2$ filter.**

**Comment 2:**

*L110: what is the rationale behind the inclusion of zonal and meridional velocities as environmental predictors for dissolved oxygen concentrations?*

**Response 2:**

Thank you to the reviewer for this question. Zonal and meridional ocean currents (east-west and north-south flows) help shape dissolved oxygen distributions by transporting water masses with different oxygen levels. For example, eastward flows like the Equatorial Undercurrent deliver oxygen-rich waters into low-oxygen zones, ventilating tropical oxygen minimum areas (Stramma et al., 2010). Similarly, lateral currents form high-oxygen "tongues" across ocean basins, showing

that advection by zonal and meridional flows is a key driver of regional oxygen patterns (Brandt et al., 2008). Including u and v velocity components as predictors thus captures the influence of ocean circulation on oxygen variability (Busecke et al., 2019).

Other studies have also included horizontal velocity fields as input features when modeling or mapping oceanic oxygen. For instance, Huang et al. (2023) incorporated reanalysis-based zonal and meridional current velocities alongside other factors to reconstruct multidecadal dissolved oxygen variability in the Indian Ocean, demonstrating that accounting for u and v improved the reconstruction by capturing lateral oxygen transport (Huang et al., 2023). Similarly, Busecke et al. (2019) found that accurately representing the Equatorial Undercurrent is essential for simulating Pacific oxygen patterns, underscoring the benefit of including current velocities in oxygen prediction models (Busecke et al., 2019).

**Comment 3:**

*L145: I understand from L111 and the documentation of ORAS5 that the data is gridded at 0.25° x 0.25° resolution. Where is this 1° x 1° grid coming from?*

**Response 3:**

ORAS5 is provided on a 0.25° mesh, but all predictors and targets in our reconstruction have to share a common $1° \times 1°$ grid for the training to be stable and for the results to be directly comparable with other products such as WOA23 and GOBAI. We therefore down-sampled the ORAS5 data by simple averaging: for every 1° cell we calculated the mean of the 16 underlying 0.25° cells that fall inside it.

**Comment 4:**

*L157: why did the authors decide to use six models in the ensemble? And why are all the algorithms tree-based? Please explain further in the text.*

**Response 4:**

Our ensemble contains six algorithms because they cover the full family of modern decision-tree learners. Random Forest represents classical bagging; Extremely Randomised Trees pushes the same idea to maximum randomness; XGBoost, LightGBM and Histogram-based Gradient Boosting all implement gradient boosting but with different split criteria and optimization tricks; CatBoost adds ordered boosting to handle categorical/ordinal inputs. Because each learner builds and regularizes its trees in a distinct way, their residual errors are only weakly correlated; averaging them therefore reduces both bias and variance.

We kept the ensemble exclusively tree-based because DO depends on non-linear interactions among various drivers, contains many missing values. All situations where decision trees excel without demanding elaborate feature engineering. We tested neural networks during preliminary experiments, but they delivered no consistent skill gain while complicating interpretability and hyper-parameter tuning. The six-tree ensemble therefore offers the best balance of predictive power and complementarity for a global DO reconstruction.

**Comment 5:**

*L164-165: why did the authors include CatBoost in the ensemble if there are no categorical features in the framework proposed (BEM-DOR)?*

**Response 5:**

We appreciate the reviewer's question. We keep CatBoost in the ensemble because its ordered-boosting scheme and symmetric trees often improve accuracy even on purely numerical data; several recent environmental studies that contained no categorical predictors at all report a clear gain from CatBoost. Chen et al. (2024) constructed an interpretable CatBoost model guided by spectral morphological characteristics for remote sensing monitoring of Chl-a and TSS along the coast of Fujian. Zhang et al. (2020) showed CatBoost outperforming generalized regression neural network (GRNN) and random forests (RF) for reference-evapotranspiration calculated from continuous meteorological variables. Because CatBoost's split selection, random strength and ordered boosting differ from our other gradient-boosting learners, its residual errors are weakly correlated with theirs. So we retained it as a complementary component of the ensemble.

**Comment 6:**

*L210: how were the hyperparameters to be tuned chosen? And how was the search range identified / selected?*

**Response 6:**

We tuned only the hyper-parameters that govern a tree model's effective capacity and regularization, because these dominate performance while the remaining options have much weaker leverage (Probst et al., 2019).   For every algorithm we therefore searched over (i) the "number of trees / iterations", which controls variance reduction; (ii) "tree depth or leaf size", which balances model complexity against over-fitting; (iii) a "learning-rate or subsampling term" to temper boosting; and (iv) the library-specific shrinkage factors such as "l2_leaf_reg" (CatBoost) or "l2_regularization" (Hist_GBT) that act as weight decay.

The hyper-parameters search ranges were chosen to coverage the values in recent large-scale environmental applications, while still being broad enough for Bayesian optimization to explore. By searching inside those empirically established intervals we avoid wasting computation on clearly unrealistic extremes while still giving Bayesian optimization enough room to find the best spot for every model.

**Comment 7:**

*L228: where would all predictions be missing? On land? Or at locations where no observations are available in the validation split during cross-validation? Please clarify further.*

**Response 7:**

The "locations where all predictions are missing" refer to ocean grid cells that contain no in-situ dissolved-oxygen profiles in any of our three source datasets (CTD, OSD, Argo) over 1960–2023, most of them lying in chronically under-sampled areas. At those cells the model still produces a concentration estimate, but in the dynamic-weighting step the local error term cannot be computed because there is no collocated observation; in that situation we replace the data-driven error with the model's prior cross-validated RMSE. Thus the grid point is retained in the final field, yet its ensemble weight is determined according to the model's historical skill rather than a point-specific mismatch. Land points are masked much earlier in the pipeline and are not part of this discussion.

**Comment 8:**

*L246-250: it is unclear to me how this temporal cross-validation differs from the cross-validation done for hyperparameter tuning. First, I would like to have a more detailed explanation of what*

*years formed the test set and what the training set, as the expression provided in line 247 is not clear. Additionally, I would like to see a detailed clarification of the differences between hyperparameter-tuning cross-validation and temporal cross-validation and the rationale behind cross-validating twice in model development.*

**Response 8:**

The eight-fold temporal cross-validation works as a sliding "leave-eight-years-out" scheme. For fold f (f = 0…7) the test set is the year sequence $\{1960 + f + 8\,k\}$ with k = 0…7; that is, eight calendar years spaced exactly eight years apart. For example, fold 0 tests on 1960 1968 1976 1984 1992 2000 2008 2016, fold 1 on 1961 1969 1977 … 2017, and so on until fold 7. All remaining years constitute the training set in that fold, so every individual year is used once for testing and seven times for training, and at no point does a model "see" future data when predicting past years.

Hyper-parameter tuning is a separate, inner procedure that we run only once. There we fix one eight-year block (1960 1968 … 2016) for training the optimizer and a different eight-year block (1967 1975 … 2023) for validation while exploring the search space. This limited split is sufficient for finding learning-rates, tree depths and similar settings   and using fewer years keeps the Bayesian search fast. After the best parameter set is locked in, we re-train each model inside the outer eight-fold loop described above.

**Comment 9:**

*L275-294 (Sect. 3.2) and then 337-366 (Sect. 4.1): Did the authors make sure that the observations they validate against in GLODAPv2 are not also included in the World Ocean Database 2023? Otherwise, they might validate against the same observations they are using to train the model. Similarly, the GOBAI-O2 product (Sharp et al., 2023) is built using GLODAPv2 observations as training data, and the product of Ito et al. (2024) is built on World Ocean Database 2018 data. How did the authors ensure that their validation data were not included in the training of these two models as well?*

**Response 9:**

We appreciate the reviewer's concern and fully agree that any overlap between GLODAP v2 and WOD23 would compromise an independent evaluation. During revision we therefore examined the two datasets profile-by-profile and did indeed find a small number of profiles that appear in both collections, as illustrated in Figure R2. Although the duplicate profiles do not always share exactly the same time-stamp or coordinates, because differences of a few kilometers or weeks are common. We adopted a deliberately conservative filter: for every oxygen profile in GLODAP v2 we searched the WOD23 CTD + OSD datasets for profiles that fall within $\pm 1°$  in longitude, $\pm 1°$  in latitude, and the same calendar month. If such a profile existed we treated the two profiles as duplicates and removed the GLODAPv2 profile from our validation pool. After applying this rule the original 56,480 GLODAPv2 profiles were reduced to 8,020. A manual spot-check confirmed that no further spatial-temporal matches remain. We therefore regard the filtered set as an independent benchmark suitable for assessing our reconstruction, GOBAI and the Ito's product.

[Figure]

**Figure R2. Identification and removal of duplicate oxygen profiles between GLODAPv2 and OSD.**
Red points mark profile locations for March 2007. a) before filtering, several GLODAPv2 profiles
(left) coincide with OSD profiles (right), indicating duplication in the two datasets. b) after
filtering, the filtered GLODAPv2 profiles (left) no longer overlaps with the remaining OSD profiles
(right), confirming that the duplicate-removal procedure successfully yields an independent
validation dataset.

Using the filtered GLODAPv2 dataset as the benchmark we repeated the comparison for all three
products; the updated numbers are summarized in Table R1. The picture hardly changes. Within
the GOBAI coverage our reconstruction still reduces RMSE by about 25 % relative to GOBAI, and
within the ITO coverage it cuts RMSE by roughly one-third compared with the Ito's product. It is
worth emphasizing that, as the reviewer notes, GLODAPv2 dataset (filtered or not) are part of
ITO's original training pool, whereas the filtered GLODAPv2 dataset were never used to train our
model. Yet the independent scores remain better for our reconstruction than for ITO itself,
confirming that the advantage stems from the dynamic weight ensemble rather than from any
inadvertent familiarity with the validation data.

Table R1. Performance comparison on the filtered GLODAPv2

| Product | MAE | RMSE | R² | ΔDO |
|---|---|---|---|---|
| Our reconstruction (full GLODAPv2) | 5.959 | 13.669 | 0.982 | 0.101 |
| GOBAI on GLODAPv2 | 11.101 | 19.875 | 0.956 | -0.971 |
| Our reconstruction in GOBAI coverage | **6.602** | **14.999** | **0.979** | **0.122** |
| ITO on GLODAPv2 | 13.415 | 22.958 | 0.951 | -0.123 |
| Our reconstruction in | **6.944** | **15.065** | **0.979** | **0.010** |

Finally, we re-evaluated the ensemble and each single model against the filtered GLODAPv2 dataset (Table R2). As expected, every single model now performs noticeably worse: their MAE rises from the earlier 5–9 µmol kg⁻¹ to 10–13 µmol kg⁻¹, RMSE rises from 11–14 to 18–20 µmol kg⁻¹, R² slips from about 0.97–0.98 to 0.96–0.967, and the ΔDO bias widens to values between −0.5 and +0.9 µmol kg⁻¹. The deterioration is consistent with the removal of inadvertent data-leakage, some of the original GLODAP profiles had been seen during training and therefore gave the single models an artificially optimistic score. The dynamic weight ensemble also performs worse, but far less dramatically, its RMSE grows by only about 3 µmol kg⁻¹ and R² remains above 0.98. This resilience confirms the value of the adaptive weighting scheme: by down-weighting locally weak learners and relying on the consensus of the remainder, the ensemble maintains superior accuracy even when the test data are entirely independent of the training set.

Table R2. Comparison of Ensemble and Single Models on the filtered GLODAPv2

| Model | MAE (µmol kg⁻¹) | RMSE (µmol kg⁻¹) | R² | ΔDO (µmol kg⁻¹) |
|---|---|---|---|---|
| Ensemble | **5.959** | **13.669** | **0.982** | 0.101 |
| Ensemble(static weight=1) | 10.327 | 18.195 | 0.967 | 0.219 |
| RF | 10.306 | 18.370 | 0.967 | **-0.011** |
| XGBoost | 11.440 | 19.420 | 0.963 | 0.323 |
| ERT | 10.668 | 18.808 | 0.965 | -0.501 |
| LightGBM | 11.215 | 18.809 | 0.965 | 0.730 |
| Hist_GBT | 12.195 | 20.030 | 0.960 | 0.361 |
| CatBoost | 12.593 | 20.072 | 0.960 | 0.963 |

**Comment 10:**

*L338: could the authors please provide a detailed description of how the comparison was performed, as it is unclear in the text?*

**Response 10:**

We thank the reviewer for asking how the comparison was performed. We first converted the GLODAPv2 data (after removing profiles duplicated in WOD23) to the grid of the product being evaluated: all observations within the cell were averaged, producing a monthly 1° × 1° field on that product's native depth levels and time span. In the first row we used the full extent of our reconstruction, global 1° grid, 75 standard depths 0-5902 m, January 1960 to December 2023, and then computed RMSE, MAE, R² and △DO between our reconstruction and the gridded GLODAPv2 field at every cell before averaging the metric over all cells. For the second row we repeated the procedure on the GOBAI grid (1° but limited to -64.5° -79.5°, 58 depths 0-2000 m, 2004-2022) and compared GOBAI to the co-located GLODAPv2 means. The third row uses the same GOBAI mask but our own reconstruction values, restricted to the depths (54 levels 0-1945 m) and period covered by GOBAI, against the corresponding GLODAPv2 cells. Similarly, the fourth row regrids GLODAPv2 onto the ITO grid (global 1°, 20 levels 0-1000 m, 1965-2020) and evaluates ITO's product; the fifth row applies the same ITO mask to our reconstruction (47 depths 0-1045 m, 1965-2020) before calculating the metrics.

**Comment 11:**

*L368: why do the authors not include the product of Roach & Bindoff (2023) in their comparison in Sects. 4.2 and 4.3, especially as that product is available up to depths of 6800 m?*

**Response 11:**

We thank the reviewer for bringing the Roach & Bindoff (2023) dataset to our attention. Our original comparison focused on products that share the same spatial and temporal resolution as ours (global 1° × 1° grids, monthly data), hence the choice of GOBAI and the Ito's product. Roach & Bindoff provide annual means only, so at the proposal stage we set it aside to avoid mixing temporal resolutions. Nevertheless, we recognize the value of their deep-reaching (to 6800 m) atlas and have now repeated the Section 4.2 and 4.3 comparisons using their data.

Figure R3 compares the 1965-2022 mean profiles down to 1000 m, while Figure R3 extends the view to 7000 m.    In the upper 50m our reconstruction and the Roach & Bindoff (RB) show almost the same curve, both lying closer to the WOA23 reference than the ITO's product. Between roughly 100 and 400 m RB fits WOA23 more tightly, whereas from 400 m to about 1000 m our line converges on the WOA23 profile and sits slightly inside the RB-WOA23 envelope.    In the intermediate and deep layers (1000-3500 m) the three products are nearly indistinguishable, but below 3500 m the RB curve remains fractionally nearer to WOA23.

[Figure]

**Figure R3. Global mean vertical profiles of different dissolved oxygen products (1965－2022).** Solid lines show our reconstruction (blue), Roach & Bindoff's reconstruction (purple), ITO's reconstruction (orange) and WOA23 climatology (yellow), plotted from the surface down to 1000 m.

[Figure]

**Figure R4. Global mean vertical profiles of different dissolved oxygen products (1965 – 2022).** Solid lines show our reconstruction (blue), Roach & Bindoff's reconstruction (purple), ITO's reconstruction (orange) and WOA23 climatology (yellow), plotted from the surface down to 7000 m.

We have also extended Section 4.3 by mapping the dissolved oxygen anomalies of the RB's dataset relative to WOA23 climatology at 10, 30, 200 and 700 m (Fig. R5). Across all four levels the RB product exhibits broader, more coherent patches of $\pm10\,\mu\text{mol kg}^{-1}$ than either our reconstruction or the ITO's product, whose departures from WOA23 remain mostly within $\pm5\,\mu\text{mol kg}^{-1}$. At present we cannot say whether these stronger spatial contrasts reflect genuine signal retained by the annual RB averaging or arise from methodological differences: our product, ITO and WOA23 are all produced on native $1° \times 1°$ grids, whereas RB's dataset results from post-processing a finer mesh into $1°$ cells by simple grid averaging, a step that may aggravate regional offsets.

[Figure]

**Figure R5. Maps of dissolved oxygen anomalies relative to WOA23 climatology at four depths (10 m, 30 m, 200 m and 700 m).** (a,c,e,g) Left panels show the difference between WOA23 and our reconstruction; (b,d,f,h) middle panels show the difference between WOA23 and ITO's reconstruction at the same depth and time period; (i,j,k,l) right panels show the difference between WOA23 and RB's dataset at the same depth and time period. White areas indicate near-zero bias, while reds and blues denote positive and negative offsets, respectively ($\pm 15$ µmol kg$^{-1}$ ).

**Comment 12:**

*L377-384: the exact difference between the lines is hard to quantify from the graph, but the authors claim that the difference between their work / WOA2023 and Ito is 2-5 umol/kg between 800-1000m when the lines seem to overlap. At the same time, they say that the difference between their study and WOA23 is 2-3 umol/kg at deeper depths down to 5902 m, while the graph clearly shows the lines diverging. This paragraph needs to be revised and the discrepancy in analytical interpretation addressed.*

**Response 12:**

We appreciate the reviewer's careful reading. We have now replaced the qualitative phrases with the exact numerical differences reported below so that the description matches the figure.

Between about 800 and 1000m the oxygen profiles are in very close agreement (Figure R3). For ITO versus WOA23 the discrepancy is 0.1 µmol kg$^{-1}$ at 800 m, 0.4 µmol kg$^{-1}$ at 900 m and 0.6 µmol kg$^{-1}$ at 1000 m. Our reconstruction diverges by 0.4 µmol kg$^{-1}$ at 857 m and becomes indistinguishable from WOA23 ($\leq$ 0.1 µmol kg$^{-1}$) at 947 m. Both products therefore track the reference climatology to well within 1 µmol kg$^{-1}$ throughout this depth range.

Below 2000m the separation grows slowly (Figure R4). From 2100 to 3900 m our field is 0.5–0.7 µmol kg$^{-1}$ lower than WOA23, with the same offset persisting to about 5000m. Deeper than 5000 m the gap widens to just over 2 µmol kg$^{-1}$, but observations are extremely sparse at those depths, so the difference mainly highlights the uncertainty of any gridded product rather

than demonstrating that either data set is definitively closer to the truth.

**Comment 13:**

*L417-429 (Sect. 5.1): this section does not add much to what is already known from a scientific perspective about large scale dissolved oxygen distribution. I suggest the authors delve deeper into some specific features of the data product that are novel compared to what is already available in the literature to provide additional evidence of why their data product is valuable. L436-464 (Sect. 5.2): similarly to the section (and comment) above, Sect. 5.2 only provides rather general and already well-known descriptions of the variations of mean dissolved oxygen concentrations throughout the water column. Additionally, the mean dissolved oxygen concentration profile in Fuigure 5 is the same as the one plohed in Figure 2.*

**Response 13:**

We thank the reviewer for highlighting the need for new science beyond a reprise of known patterns, in response we have added Sections 5.3 and 5.4.    Section 5.3 presents a hemispheric seasonal analysis (Fig. R6).    In the surface layer (0–100 m) the phase-opposed seasonal cycle in the two hemispheres reflects the WOA23 reference almost exactly, demonstrating that the ensemble retains the typical ventilation signal without relying on WOA as a predictor.    Because the reconstruction extends to 5902 m, we can for the first time quantify the seasonal amplitude of deep-ocean oxygen: it is everywhere below 0.1 µmol kg⁻¹ between 1516 and 5902 m, a depth range not covered by GOBAI or ITO.    This virtual absence of seasonality provides a new observational constraint on abyssal ventilation and on biogeochemical modelling of tropical and polar deep waters.

[Figure]

**Figure R6. Hemispheric mean climatological seasonal cycle of dissolved-oxygen anomalies.** Monthly anomalies are averaged over the Northern Hemisphere (left) and Southern Hemisphere (right). Solid curves show the present reconstruction for three depth ranges: 0‒97 m (blue), 97‒1516 m (orange), and 1516‒5920 m (yellow). Dashed curves give the WOA23 climatology for the 0‒100 m and 100‒1500 m.

Figure R7 presents January and July sections along 160° W.    The mean panels highlight the classic pattern of high-oxygen polar waters, a broad equatorial minimum and weak vertical gradients below about 2000m.    The trend panels reveal a coherent de-oxygenation signal confined to the upper 1000m: losses reach –0.5 µmol kg⁻¹ yr⁻¹ in the sub-polar Southern Ocean

and in the mid-latitude North Pacific, while the equatorial region shows only weak changes. Below roughly 2000 m the trends collapse to near-zero, indicating that deep Pacific oxygen has remained effectively stable over 1960–2023.    Comparison of January and July sections shows no systematic shift in the sign or magnitude of the trends, suggesting that long-term changes dominate and any seasonality is secondary along this longitude.

[Figure]

**Figure R7. Latitude–depth sections of dissolved oxygen at 160 °W.** Left column: climatological mean concentration (μmol kg⁻¹) for January (top) and July (bottom). Right column: linear trend over 1960–2023 (μmol kg⁻¹ yr⁻¹) for the same months; grey X marks denote grid cells where the trend is not significant at the 95 % level.

Figure R8 shows the January and July sections along 30 °W, the approximate mid-Atlantic meridian.    A pronounced oxygen increase of +0.3 to +0.5 μmol kg⁻¹ yr⁻¹ occupies the upper 1200 m of the mid-latitude North Atlantic (40–60 °N).    South of about 40 °S the upper 1000m exhibits a broad de-oxygenation region, −0.1 to −0.3 μmol kg⁻¹ yr⁻¹.    As in the Pacific section, trends below 2000 m are essentially zero, indicating that abyssal Atlantic oxygen has remained stable from 1960 to 2023.    The January and July panels are nearly identical, just like the Pacific section.

[Figure]

**Figure R8. Latitude–depth sections of dissolved oxygen at 30 °W.** Left column: climatological mean concentration (μmol kg⁻¹) for January (top) and July (bottom). Right column: linear trend over 1960–2023 (μmol kg⁻¹ yr⁻¹) for the same months; grey X marks denote grid cells where the trend is not significant at the 95 % level.

**Comment 14:**

*Dataset in netcdf format: the values of 'time' and 'depth' seem to be decoded incorrectly in*

*the final version of the file. Time is only reported as timesteps (0 to 767; without any decodable information on month or year). When opening the file in 'ncview', depth is only readable as depth level (1 to 74, without any information on the depth value in meters). Lastly, latitude, longitude and depth are included in the dataset as variables instead of coordinates.*

**Response 14:**

Time is a coordinate variable (its name matches the dimension) with the units "months since 1960-01-01 00:00:00". The numeric values therefore run 0 … 767. Depth is likewise a coordinate variable with units m; the values are the true layer depths (0 m, 10 m, … 5902 m).    Latitude and longitude appear under "Variables" because netCDF lists every 1-D coordinate there, but their names equal the dimensions (lat, lon), so they are valid CF coordinate variables. In short, the coordinates are correctly encoded; the issue is a viewer limitation rather than a decoding error. For reproducibility, here is a minimal MATLAB snippet that reads the file:

ncdisp('DO_Reconstruction_1960-2023.nc')

ncinfo('DO_Reconstruction_1960-2023.nc')

DO_Reconstruction = ncread('DO_Reconstruction_1960-2023.nc','DO_Reconstruction');

Longitude = ncread('DO_Reconstruction_1960-2023.nc','Longitude');

Latitude = ncread('DO_Reconstruction_1960-2023.nc','Latitude');

Depth = ncread('DO_Reconstruction_1960-2023.nc','Depth');

*Technical corrections*

**Response 15:**

Thank you for the careful technical corrections. We have implemented every suggested change and double-checked the manuscript. Your detailed remarks have improved the clarity and accuracy of the manuscript.

**Reference**

Brandt P, Hormann V, Bourlès B, et al. Oxygen tongues and zonal currents in the equatorial Atlantic. *Journal of Geophysical Research: Oceans*, **2008**, 113(C4).

Busecke J J M, Resplandy L, Dunne J P. The equatorial undercurrent and the oxygen minimum zone in the Pacific. *Geophysical Research Letters*, **2019**, 46(12): 6716-6725.

Chen B, Chen Y, Chen H. An interpretable CatBoost model guided by spectral morphological features for the inversion of coastal water quality parameters. *Water*, **2024**, 16(24): 3615.

Huang S, Shao J, Chen Y, et al. Reconstruction of dissolved oxygen in the Indian Ocean from 1980 to 2019 based on machine learning techniques. *Frontiers in Marine Science*, **2023**, 10: 1291232.

Probst P, Boulesteix A L, Bischl B. Tunability: Importance of hyperparameters of machine learning algorithms. *Journal of Machine Learning Research*, **2019**, 20(53): 1-32.

Stramma L, Schmidtko S, Levin L A, et al. Ocean oxygen minima expansions and their biological impacts. *Deep Sea Research Part I: Oceanographic Research Papers*, **2010**, 57(4): 587-595.

Zhang Y, Zhao Z, Zheng J. CatBoost: A new approach for estimating daily reference crop evapotranspiration in arid and semi-arid regions of Northern China. *Journal of Hydrology*, **2020**, 588: 125087.